# Exploring Discourse Structure in Document-level Machine Translation

**Xinyu Hu** and **Xiaojun Wan**
Wangxuan Institute of Computer Technology, Peking University
{huxinyu,wanxiaojun}@pku.edu.cn

## Abstract

Neural machine translation has achieved great success in the past few years with the help of transformer architectures and large-scale bilingual corpora. However, when the source text gradually grows into an entire document, the performance of current methods for document-level machine translation (DocMT) is less satisfactory. Although the context is beneficial to the translation in general, it is difficult for traditional methods to utilize such long-range information. Previous studies on DocMT have concentrated on extra contents such as multiple surrounding sentences and input instances divided by a fixed length. We suppose that they ignore the structure inside the source text, which leads to under-utilization of the context. In this paper, we present a more sound paragraph-to-paragraph translation mode and explore whether discourse structure can improve DocMT. We introduce several methods from different perspectives, among which our RST-Att model with a multi-granularity attention mechanism based on the RST parsing tree works best. The experiments show that our method indeed utilizes discourse information and performs better than previous work.

## 1 Introduction

Transformer (Vaswani et al., 2017) based approaches, together with adequate bilingual datasets, have led to significant progress on machine translation (MT). However, the performance of MT models usually drops dramatically when processing long texts. Although document-level machine translation (DocMT) can be solved with sentence-level MT by translating each sentence separately, the potential information in the long-range context may be ignored. To address these problems, many methods in DocMT have been proposed to better utilize the contextual information and improve the overall translation quality of the document.

Among these methods, the dominant approaches still adhere to the sentence-by-sentence mode, but they utilize additional contextual information, including the surrounding sentences (Zhang et al., 2018; Miculicich et al., 2018; Kang et al., 2020; Zhang et al., 2020b, 2021a), document contextual representation (Jiang et al., 2020; Ma et al., 2020) and memory units (Feng et al., 2022). In recent years, many researches have turned to translating multiple sentences or the entire document at once (Tan et al., 2019; Bao et al., 2021; Sun et al., 2022; Li et al., 2022). However, previous work (Zhang et al., 2018; Liu et al., 2020; Sun et al., 2022) has demonstrated that direct Doc2Doc translation may cause the model not to converge. Therefore, those methods adopt pertinent measures such as data augmentation, text truncation, and specific frameworks for sentence alignment.

Despite the effectiveness of previous efforts on DocMT, some research, such as Kim et al. (2019), suggested that existing methods may not fully utilize the context and the improvement may come from regularization. Kang et al. (2020) also indicated that dynamic context selected from the surrounding sentences can improve translation more efficiently. The additional contexts used in most previous work are simply based on the distance from the current sentence or the length of the text, which is somewhat arbitrary. In this paper, we aim to explore a more reasonable way to encode context through the discourse structure.

Discourse structure refers to how elementary text units are organized to form a discourse and logically linked to one another. Early studies have demonstrated that discourse parsing can benefit various downstream NLP tasks, including sentiment analysis (Bhatia et al., 2015), relation extraction (Wang et al., 2021), text summarization (Gerani et al., 2014; Xu et al., 2020), machine translation evaluation (Guzmán et al., 2014; Joty et al., 2014, 2017; Bawden et al., 2018) and so on. RST parsing, based on Rhetorical Structure Theory (Mann and Thompson, 1987), is one of

the most influential parsing methods in discourse analysis. According to RST, a text is segmented into several clause-like units (EDUs) as leaves of the corresponding parsing tree. Through certain rhetorical relations among adjacent spans, underlying EDUs or larger text spans are recursively linked and merged to form their parent nodes, representing the concatenation of them.

Although RST parsing would be better conducted on an integral text to maintain discourse structure, existing models perform poorly on long texts. As a result, we present a new paragraph-to-paragraph translation mode, where the original document is divided into several shorter paragraphs. Our paragraph segmentation is generated by the TextTiling tool (Hearst, 1997) based on subtopic shifts and discourse cues, since frequently-used datasets of DocMT do not contain paragraph alignment tags. We suppose the discourse analysis of paragraphs is a proper compromise and more sound than previous partitioning methods. And it is more labor-saving when dealing with other multilingual corpora equipped with paragraph alignments.

We employ the end-to-end method of Hu and Wan (2023) to train our RST parsing model. The parsing task is reformulated into a Seq2Seq task through a linearization process and then trained based on a pretrained language model. Several attempts have been made from different perspectives to investigate how to utilize the discourse structure, including the integration of the RST sequence, an RST-specific attention mechanism, and graph learning based on the RST tree. Among them, our RST-Att model, designed with a multi-granularity attention mechanism to inject the RST tree into the translation encoder, achieves the best performance. Specifically, each token at the first encoder layer can only notice other tokens from the EDU it belongs to, when calculating self-attention scores. As the encoder layer moves backward, the range of attention continuously expands to complete context according to the structure of the RST tree. We believe such a progressive pattern can reduce the difficulty of modeling long-range context.

Overall, our main contributions are as follows:

1) We generate the paragraph segmentation and introduce a more sound paragraph-to-paragraph translation mode than traditional text partition.

2) We explore several methods to take advantage of the discourse structure predicted by the RST

parsing model to improve document-level machine translation.

3) Our RST-Att model achieves superior performance on three widely used datasets of DocMT and further linguistic evaluation, compared with existing works.[1]

## 2 Methodology

In this section, we will elaborate on the relevant steps to obtain the discourse structure information and explore its utilization in DocMT. In Section 2.1, we compare two text segmentation methods and determine the TextTiling tool for our paragraph segmentation according to the distribution of results. In Section 2.2, we introduce the brief training process of the RST parsing model and how to linearize the corresponding RST tree. Section 2.3 describes the construction and details of our proposed RST sequence integration method, as well as the RST-Att model with the RST-specific attention mechanism of multiple levels of granularity.

### 2.1 Paragraph Segmentation

In this step, we consider two distinct approaches for paragraph segmentation. The first is the traditional TextTiling method, which detects topic and lexical co-occurrences. The other is a neural segmentation model proposed in recent years by Lukasik et al. (2020), based on a pretrained language model and cross-segment attention.

**TextTiling Algorithm** TextTiling is an algorithm for tokenizing a document into multi-paragraph units that represent subtopics, proposed by Hearst (1997). The algorithm identifies subtopic shifts based on the discourse cues of lexical co-occurrence and distribution patterns. It first divides the whole document into several parts, each of which contains multiple unbroken sentences. Then the similarity scores are calculated between sentences, and the peak differences between them are found to become the final segmentation boundaries after the normalization.

**Neural Segmentation Model** Lukasik et al. (2020) proposed a neural model with transformer-based architectures. They represented each candidate break using its left and right local contexts and then turned to the pretrained language model

---
[1]Codes and data are available at https://github.com/herrxy/RST-DocMT.

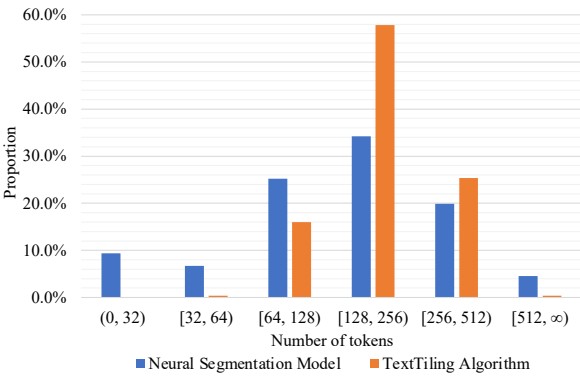

Figure 1: The distribution of the number of tokens in each divided paragraph.

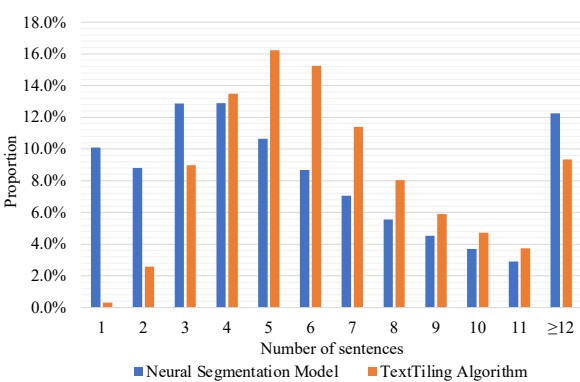

Figure 2: The distribution of the number of sentences in each divided paragraph.

for judging whether the candidate break was reasonable. The LM was finetuned on a large-scale dataset, Wiki-727K (Koshorek et al., 2018).

We apply the two methods above to the most commonly used dataset of document-level machine translation and obtain the respective paragraph segmentation. We take the dataset News as an example, and the distributions of the number of tokens and sentences contained in each paragraph are shown in Figure 1 and Figure 2. The vertical axis of the legends shows the proportion of the corresponding category in all samples. The paragraphs from the TextTiling method have a more reasonable distribution, and most of them contain moderate numbers of tokens and sentences. On the other hand, a considerable part of the paragraphs obtained by the neural segmentation model only contain a few sentences, which is not conducive to subsequent discourse parsing. Therefore, we finally choose the results of the TextTiling method for the following experiments, and the statistical details of paragraph segmentation can be found in Table 1.

## 2.2 RST Parsing

Previous studies have proposed many methods for RST parsing (Lin et al., 2019; Zhang et al., 2020a; Kobayashi et al., 2020; Nguyen et al., 2021). However, most of them split the parsing process into two steps: EDU segmentation and RST tree prediction, for which the gold EDU labels are often required. Considering that the datasets of DocMT are not equipped with such information, we follow Hu and Wan (2023) to train an end-to-end RST parser from scratch through a Seq2Seq reformulation method. The training data comes from the standard RST Discourse TreeBank (Carlson et al., 2001) and has been processed with a similar length distribution to our paragraph segmentation.

**Linearization** Based on the priority of brackets, we represent hierarchical architecture by nesting several pairs of brackets. The linearization is carried out from the bottom up, according to post-order traversal. We replace each leaf that represents a single EDU with a sequence comprised of a left bracket, text content, a right bracket, and its nuclearity and rhetorical relation labels. The same process is performed for other nodes, except that the concatenation of new representations of two child nodes serves as the text content. The linearized sequence is designed to contain the complete original input text for better performance, according to the observation of Paolini et al. (2021). More details can be found in Hu and Wan (2023), and an example is shown in Figure 3(d).

**Seq2Seq Training** Since the input and new output of the task are both sequences, we have trained our RST parsing model on the pretrained Seq2Seq model T5-base (Raffel et al., 2020). The related latent knowledge may be transferred and utilized during training since the reformulated sequences are close to natural language text, which aligns with the pretraining of T5. Moreover, we modify and align the output predicted by the model with the format we design before to obtain the final RST parsing tree through a recursive algorithm.

## 2.3 Utilizing RST Structure

### 2.3.1 RST Sequence Integration

We first attempt a simple method that directly integrates the discourse parsing tree into the inputs of the model, called RST-Seq. The source input during training is replaced with the corresponding linearized RST tree, and the target output is

**(a) Input Text**

Government lending was not intended to be a way to obfuscate spending figures, hide fraudulent activity, or provide large subsidies.

**(b) EDU Segmentation**

$EDU_1$: Government lending was not intended to be a way

$EDU_2$: to obfuscate spending figures,

$EDU_3$: hide fraudulent activity,

$EDU_4$: or provide large subsidies.

**(c) RST Parsing Tree**

Root

(Nucleus, Span)   (Satellite, Elaboration)

(Nucleus, joint)   (Nucleus, joint)

(Nucleus, joint)   (Nucleus, joint)

$EDU_1$   $EDU_2$   $EDU_3$   $EDU_4$

**(d) Linearization**

[ [ Government lending was not intended to be a way ] Nucleus span [ [ to obfuscate spending figures, ] Nucleus joint [ [ hide fraudulent activity, ] Nucleus joint [ or provide large subsidies. ] Nucleus joint ] Nucleus joint ] Satellite elaboration ]

Figure 3: A brief example from RST Discourse TreeBank, with the EDU segmentation (part b), the RST parsing tree of the input text (part c), and the corresponding sequence of the linearized RST tree (part d).

kept the same. The experiments have been made on the well-trained mBART25 (Liu et al., 2020), similar to the previous work. We expect that the pretrained language model can encode the information of discourse structure together with the context through latent knowledge. The results of our RST-Seq method and the baseline without the discourse structure will be shown in Section 4.

### 2.3.2 Multi-Granularity Attention

To further take advantage of the discourse structure, we propose the RST-Att model with a multi-granularity attention mechanism based on the RST tree, inspired by Wu et al. (2018). From the first layer of the encoder to the last, the range that each token can attend to continuously expands according to the bottom-up nodes of the RST parsing tree. According to Beltagy et al. (2020), context may be better integrated in the higher layer, whereas the lower layer tends to encode local information. We suppose the model will better understand the source context in a progressive way under the instruction of discourse structure.

Specifically, we first transform the RST parsing tree, combining the leaf nodes and their parent nodes. Then each node in the tree is assigned its height, which refers to the number of edges on its longest path to a leaf node. We ignore the label information and replace each node with the range of EDUs it represents, as shown in Figure 4. Obviously, each leaf node has a height of 0, and only the root has the largest height. We construct the initial node set $S_l$ which consists of all nodes with heights no more than $l$. Then we delete the node from $S_l$ if it is a descendant node of another node in $S_l$, until

the set will not change. It can be simply proved that the text ranges represented by the nodes of finally obtained set $\hat{S}_l$ perfectly cover the entire paragraph content. We assume that $\hat{S}_l = (r_1, r_2, \cdots, r_{n_l})$, where the node $r_k$ covers the token range from the position of $k_{\text{begin}}$ to $k_{\text{end}}$ in the paragraph.

In the encoder layer $l$ of the original transformer model, the multihead self-attention is computed as:

$$A^l = \text{MultiHead}(\text{Softmax}(\frac{Q_l K_l^T}{\sqrt{d}}))$$

where $Q_l$ and $K_l$ are the query and key matrices with the vector dimension of $d$ in the $l$ th layer. We then modify the calculation of the attention matrix $A^l$ to $\hat{A}^l$ according to the text ranges in $\hat{S}_l$:

$$\hat{A}^l = \text{MultiHead}(\text{Softmax}(\frac{Q_l K_l^T}{\sqrt{d}} + M^l))$$

$$M_{ij}^l = \begin{cases} 0 & \exists\, r_k \in \hat{S}_l,\ k_{\text{begin}} \le i, j \le k_{\text{end}} \\ -\infty & else \end{cases}$$

where the matrix $M^l$ has the size of $N \times N$ and $N$ presents the number of tokens in the paragraph.

Through the modified attention mechanism, different granularities for context modeling can be implemented by different encoder layers with specific attention ranges of tokens. Specifically, at the first layer, each token can only notice other tokens from the EDU that it belongs to. As for the last layer, each token can attend to all tokens in the paragraph, which turns back to the original full attention mode. We simulate as many levels of granularity as possible, and our method is illustrated with a simple example in Figure 4. The RST-Att model

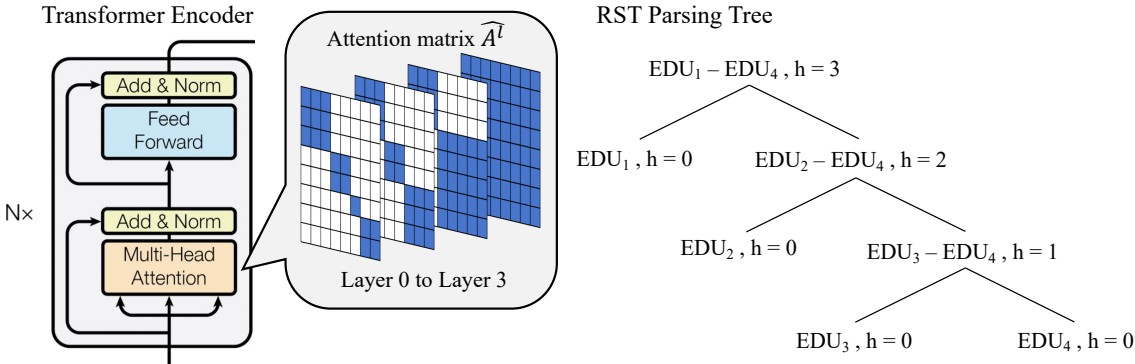

Layer 0: [ Government lending was not intended to be a way ] $_{EDU_1}$ [ to obfuscate spending figures, ] $_{EDU_2}$ [ hide fraudulent activity, ] $_{EDU_3}$ [ or provide large subsidies. ] $_{EDU_4}$

Layer 1: [ Government lending was not intended to be a way ] $_{EDU_1}$ [ to obfuscate spending figures, ] $_{EDU_2}$ [ hide fraudulent activity, or provide large subsidies. ] $_{EDU_3-EDU_4}$

Layer 2: [ Government lending was not intended to be a way ] $_{EDU_1}$ [ to obfuscate spending figures, hide fraudulent activity, or provide large subsidies. ] $_{EDU_2-EDU_4}$

Layer 3: [ Government lending was not intended to be a way to obfuscate spending figures, hide fraudulent activity, or provide large subsidies. ] $_{EDU_1-EDU_4}$

Figure 4: The illustration of our RST-Att model with a simple example whose RST parsing tree has a height of 3, mapping to only four encoder layers. The multi-head attention of the vanilla transformer encoder (Vaswani et al., 2017) is modified for different layers.

does not introduce additional parameters; instead, it theoretically reduces computational overhead and improves efficiency with incomplete attention.

Moreover, it should be mentioned that the encoder contains a fixed number of layers, while the heights of RST trees vary. So we should construct a mapping to guarantee each encoder layer is linked to a certain node set. To keep the mapping uniform, the new set $\tilde{S}_i$ is calculated as follows:

$$\tilde{S}_i = \hat{S_{m_i}}, \quad m_i = \lfloor \frac{H}{L-1} i \rfloor$$

where $i = 0, 1, \cdots, L-1$ and $L, H$ denote the number of encoder layers and the height of the RST tree respectively. And $m_i$ refers to the old layer index to be mapped.

## 3 Datasets and Settings

We evaluate our models on three widely used datasets for document-level machine translation for English to German, from Maruf et al. (2019).

**TED** The corpus includes TED talks and corresponding translations from IWSLT 2017. tst2016-2017 is used as the test set and the rest as valid.

**News** The corpus comes from News Commentary v11. Newstest2016 is used as the test set and newstest2015 as valid.

**Europarl** The corpus is extracted from Europarl v7, which is split into the training, test, and valid sets, as mentioned in Maruf et al. (2019).

The detailed statistics of these datasets and their paragraph segmentation are displayed in Table 1. Similar to the previous work, Moses (Koehn et al., 2007) is used for data processing and sentence truecase. We apply BPE (Sennrich et al., 2016) with 32K merge operations on both sides for all datasets.

Following previous work (Bao et al., 2021; Li et al., 2022; Sun et al., 2022; Feng et al., 2022), we apply sentence-level BLEU score (s-BLEU) and document-level BLEU score (d-BLEU) as the metrics of evaluation. Since our methods are focused on the DocMT and do not involve sentence alignments, the d-BLEU score is our major metric, which matches n-grams in the whole document.

Since our method can be directly applied to pre-trained language models that have been commonly employed in current research, we conduct our experiments based on mBART25 (Liu et al., 2020). It is a strong multilingual model with the transformer architecture and contains about 610M parameters. The setting aligns with the previous state-of-the-art model G-Transformer (Bao et al., 2021), which serves as our primary comparison objective. More experiment details are described in Appendix A.

| Dataset | #Sentences | #Documents | #Paragraphs | Avg #Tokens/Para | Avg #Sents/Para |
|---|---|---|---|---|---|
| TED | 0.21M/9K/2.3K | 1.7K/93/23 | 23K/966/230 | 217/217/232 | 9.0/9.3/9.9 |
| News | 0.24M/2K/3K | 6.1K/71/155 | 35K/296/429 | 212/200/189 | 6.7/7.3/7.0 |
| Europarl | 1.67M/3.6K/5.1K | 118K/240/360 | 318K/691/988 | 169/170/170 | 5.3/5.2/5.2 |

Table 1: The detailed statistics of the datasets used for DocMT in the form of train/vaild/test.

| Model | TED | | News | | Europarl | |
|---|---|---|---|---|---|---|
| | s-BLEU | d-BLEU | s-BLEU | d-BLEU | s-BLEU | d-BLEU |
| Sent2Sent (Sun et al., 2022) | 25.19 | 29.16 | 24.98 | 27.03 | 31.70 | 33.83 |
| MCN (Zheng et al., 2020) | 25.10 | 29.09 | 24.91 | 26.97 | 30.40 | 32.63 |
| G-Transformer (Bao et al., 2021) | 25.12 | 27.17 | 25.52 | 27.11 | 32.39 | 34.08 |
| MR Doc2Sent (Sun et al., 2022) | 25.24 | 29.20 | 25.00 | 26.70 | 32.11 | 34.18 |
| MR Doc2Doc (Sun et al., 2022)* | - | 29.27 | - | 26.71 | - | 34.48 |
| Recurrent Memory (Feng et al., 2022) | 25.62 | 29.47 | 25.73 | 27.78 | 31.41 | 33.50 |
| P-Transformer (Li et al., 2022) | 25.67 | 27.94 | 25.93 | 27.67 | 32.62 | 34.49 |
| Based on pretrained language models | | | | | | |
| Flat-Transformer (BERT) (Ma et al., 2020) | 26.61 | - | 24.52 | - | 31.99 | - |
| SDoc2SDoc (BART) (Bao et al., 2021) | - | 28.29 | - | 30.49 | - | 34.00 |
| G-Transformer (BART) (Bao et al., 2021) | 28.06 | 30.03 | 30.34 | 31.71 | 32.74 | 34.31 |
| Para2Para Baseline (BART) | - | 30.35 | - | 31.43 | - | 34.19 |
| RST-Seq (BART) | - | 30.47 | - | 31.62 | - | 34.22 |
| RST-Att (BART) | - | **31.10** | - | **32.28** | - | **34.55** |

Table 2: Results of our models and previous work on document-level machine translation. *Although Sun et al. (2022) achieved better results with additional sentence corpus, we do not consider them here for fairness.

## 4  Experiments and Analysis

We compare the experimental results of our methods with previous approaches to document-level machine translation on TED, News, and Europarl, as shown in Table 2. We choose the best result of Sent2Sent here from Sun et al. (2022), which even surpasses some DocMT models on d-BLEU, indicating the underuse of context in their work. Moreover, some earlier studies are ignored since they only reported s-BLEU scores and performed much worse than recent work such as G-Transformer. And we categorize all methods based on whether they are based on pretrained language models.

Above all, we introduce our Para2Para baseline, which is finetuned on the original mBART25 model. It simply performs paragraph-to-paragraph translation based on the segmented paragraphs without utilizing any RST discourse information. For better comparison, we also present the results of the traditional sub-document baseline SDoc2SDoc from Bao et al. (2021). As described in Section 1, many studies on DocMT have adopted splitting each document into sub-documents for translation

based on a specified maximum length, since standard Doc2Doc translation was pointed out to be prone to training failures.

The results show that our Para2Para baseline outperforms the traditional SDoc2SDoc with the same mBART25 model, proving the improvement of our new translation mode. It can be attributed to the fact that simple SDoc2SDoc mode, without any other optimization, may arbitrarily ignore the interactions between strongly related sentences. In contrast, our proposed paragraph-to-paragraph method takes them into consideration and is expected to make better use of contextual information.

Furthermore, we introduce discourse information and propose our RST-Seq and RST-Att models. Both of them follow the Para2Para translation mode based on mBART25, and contain the same number of parameters as the baselines mentioned above. Although RST-Seq performs better than the Para2Para baseline, the improvements are not prominent. We suppose that much longer inputs of linearized RST trees may increase the difficulty of training. Moreover, it may be difficult for mBART25 to directly adapt to the format of the RST sequence since it

| Model | TED | News |
|---|---|---|
| RST-Att | 31.10 | 32.28 |
| Six-level granularity | 30.80 (-0.30) | 32.04 (-0.24) |
| Three-level granularity | 30.75 (-0.35) | 31.78 (-0.50) |
| Equal division | 30.31 (-0.79) | 31.55 (-0.73) |

Table 3: The d-BLEU scores of the experiments in granularity discussion with different settings.

has not been finetuned for such parsing tasks and there are not enough hints in the translation task.

In addition, our RST-Att model achieves the best results and improves d-BLEU scores by 1.07, 0.57, and 0.06 on TED, News, and Europarl, respectively, showing the effectiveness of our proposed attention mechanism. Its superiority over the RST-Seq demonstrates that discourse knowledge may be applied to the model structure more effectively than the input data. Furthermore, Table 1 shows that the data scales of TED, News, and Europarl are increasing while the improvements on the corresponding dataset are decreasing. We believe that our RST-specific attention module can alleviate the negative impact of small datasets to a certain extent. When the training data is sufficient, like in the case of Europarl, the performance gaps between all existing models are not significant.

### 4.1 Granularity Discussion

To demonstrate the effectiveness of our introduction of the RST discourse structure, we evaluate the RST-Att model with different settings of the involved multi-granularity attention mechanism.

**Different Number of Levels**  We first verify whether using as much granularity as possible can improve the performance of the model. The node sets are instead mapped to just six encoder layers, corresponding to half of the original levels of granularity. We make a copy of each mapped layer to fill the whole encoder progressively. Furthermore, we attempt the case of much less granularity in a clearer manner of partitioning, including three levels of EDU, sentence and paragraph. And similarly, each layer of these three levels is repeated four times after mapping.

**Equally Divided Tree**  We also pay attention to the impact of discourse guidance in our multi-granularity attention module. To exclude the discourse information, we conduct a more direct implementation of multiple granularity based on the

| Model | deixis | lex.c | ell.infl | ell.VP |
|---|---|---|---|---|
| Sent2Sent | 50.0 | 45.9 | 53.0 | 28.9 |
| Concat | 83.5 | 47.5 | 76.6 | 76.2 |
| CADec | 81.6 | 58.1 | 72.2 | 80.0 |
| LSTM-Transformer | **91.0** | 46.9 | 82.2 | 78.2 |
| MR-Doc2Doc | 64.7 | 46.3 | 65.9 | 53.0 |
| G-Transformer | 89.9 | - | 84.8 | 82.4 |
| ChatGPT | 57.9 | 44.4 | 75.0 | 71.6 |
| GPT-4 | 85.9 | 72.4 | 69.8 | 81.4 |
| RST-Att (ours) | 87.2 | **81.7** | **85.2** | **87.2** |

Table 4: Accuracy [%] of translation prediction for specific discourse phenomena (deixis, lexical consistency, ellipsis of morphological inflection, and VP ellipsis) among different models on the contrastive test sets.

equally divided tree. Specifically, each range of content is divided into two equal ones, layer by layer, until the current content contains no more than three tokens.[2]

The results are shown in Table 3. The drops in performance become larger from the model with six-level granularity to the model with three-level granularity, proving that more elaborate granularity levels contribute to the improvement of the RST-Att model. On the other hand, despite also containing multiple granularity, the model constructed through arbitrarily equal division gets even worse results, which further demonstrates the crucial and important role of discourse structure in our method.

### 4.2 Linguistic Evaluation

Furthermore, we have conducted linguistic evaluations about the performance of models when dealing with discourse phenomena, based on the frequently used contrastive test sets (Voita et al., 2019) in existing works. They were designed for targeted evaluation of several discourse phenomena: deixis, ellipsis (two test sets for VP ellipsis and morphological inflection), and lexical cohesion. Each test instance consists of a true example and several contrastive translations that differ from the true one only in the considered discourse aspect. The translation system needs to select the one that it considers to be true, which is evaluated by accuracy. We compared the results of Sent2Sent, Concat and CADec from Voita et al. (2019), LSTM-Transformer from Zhang et al. (2020b), G-Transformer, and MR-Doc2Doc, Chat-

---

[2]Smaller text units will cause the failure of training according to our experiments.

GPT[3] and GPT-4 from Wang et al. (2023) with our RST-Att model, as shown in Table 4.

Our RST-Att model has achieved superior performance, with the highest accuracy in three aspects: lexical cohesion, VP ellipsis, and morphological inflection, while having lower performance in deixis compared to SOTA. And the improvement in terms of lexical cohesion over previous methods is notably significant. We suppose it may be because the RST discourse structure that pays attention to the relationship between text units can promote the model to better handle lexical cohesion. Moreover, our approach outperforms LLMs comprehensively, including ChatGPT and GPT-4, and more discussions with LLM are described in Appendix B. These results indicate the enhanced ability of our model to adapt to contextual information and handle discourse phenomena, which shows the promising way to introduce discourse information into document-level machine translation.

### 4.3 Context Length

Next, we compare our RST-Att model with the baseline with respect to different lengths of input context. Since our translation is based on paragraphs, we follow the d-BLEU to calculate the p-BLEU score, which matches the n-grams in each paragraph. Figure 5 depicts the results of two models on News. Surprisingly, the baseline, whose structure is still the vanilla transformer, does not fail on long contexts, contradicting the findings of Li et al. (2022). We consider that the difference may be due to the knowledge of the well-trained LM. Moreover, the result of our RST-Att model exceeds the baseline at each length, which is more distinct as the length increases. And it maintains a relatively stable performance when the input length is more than 256, indicating the advancement of our model in dealing with long inputs.

### 4.4 Label Utilization

Since the RST-Att model ignores nuclearity and relations on the RST parsing tree, we have further explored whether the in-depth utilization of label information can lead to more improvements. We apply graph learning to enhance the comprehension of the model for each EDU. RST parsing trees are transformed into dependency trees according to Hirao et al. (2013), so that each node will represent a different EDU. We serve the dependency

---

[3]https://chat.openai.com.

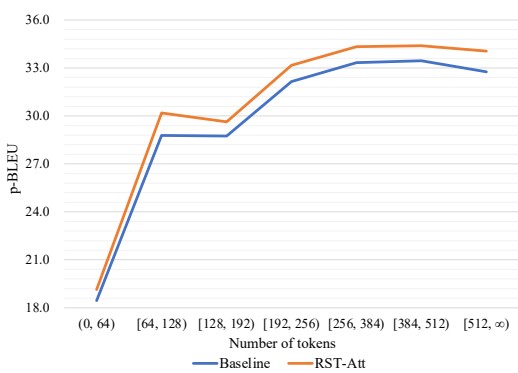

Figure 5: The p-BLEU scores of the baseline and our RST-Att model on different input lengths.

tree as a graph, and the connectivity and the path to other nodes with label sequences can be calculated for each EDU. These features are encoded to update the representation of each EDU with the architecture of GraphFormer (Yang et al., 2021). Then we integrate the learned representation into the calculation of each token the EDU contains.

Although this method surpasses the baseline, there is no significant improvement over the RST-Att model. We suppose that the path with label information can represent the relationship between nodes at both ends, but it may be too complicated for the model to encode such knowledge. On the other hand, there are still many errors in the predicted labels on account of the limitations of current RST parsing research, which may mislead the model during training. We hope our exploration can inspire future research to come up with more effective approaches to utilizing RST labels.

## 5 Related Works

**Document-level Machine Translation** Although early work has achieved great success on machine translation, a document is often processed by translating each sentence separately. As a result, the information included in context is ignored. Recently, document-level machine translation has attracted more attention and many methods have been proposed to better utilize the contextual information to improve translation quality.

Most early attempts still followed sentence-to-sentence translation, but they applied various frameworks to utilize the context during training. Zhang et al. (2018); Miculicich et al. (2018); Zhang et al. (2020b); Zheng et al. (2020); Zhang et al. (2021a) utilized surrounding sentences and integrated the contextual information into encoding of the current

sentence. Kang et al. (2020) indicated dynamic context selected from the surrounding sentences can improve the quality of the translation more effectively. Jiang et al. (2020); Ma et al. (2020) designed the specific module to encode the document into a contextual representation. Feng et al. (2022) introduced a recurrent memory unit to remember the information thorough the whole document.

Although Zhang et al. (2018); Liu et al. (2020) have shown that direct Doc2Doc translation may cause the failure of training, many recent studies have focused on translating multiple sentences or the entire document at once (Tan et al., 2019; Bao et al., 2021; Sun et al., 2022; Li et al., 2022). They used various specific methods to avoid this problem and achieved more advancement on DocMT.

**Discourse Parsing** Discourse parsing describes the hierarchical tree structure of a text and can be used in quality evaluations like coherence and other downstream applications. RST parsing is the most important role of discourse parsing, and the existing approaches can be mainly divided into two classes: top-down and bottom-up paradigms.

Bottom-up methods have been first proposed since hand-engineered features were suitable for representing local information. Models with CKY-like algorithms (Hernault et al., 2010; Joty et al., 2013; Feng and Hirst, 2014; Li et al., 2014) utilized diverse features to learn the scores for candidate trees and selected the most possible one. Another common bottom-up method is the transition-based parser with actions of shift and reduce (Ji and Eisenstein, 2014; Wang et al., 2017; Yu et al., 2018).

Recent advancements in neural methods made global representation more effective, which promoted top-down parsers. Lin et al. (2019) first presented a Seq2Seq model based on pointer networks (Vinyals et al., 2015) and Liu et al. (2019) improved it with hierarchical structure. Then Zhang et al. (2020a) extended their methods to document-level RST parsing. Kobayashi et al. (2020) constructed subtrees with three levels of granularity and merged them together.

Despite the better performance of top-down models, most of them still need gold EDU segmentation and drop a lot in performance when using automatic segmenters. To address the problem, Nguyen et al. (2021) introduced an end-to-end parsing model, relying on specific frameworks for different tasks. Zhang et al. (2021b) proposed a system with rerankers to improve the performance.

## 6 Conclusions

In this paper, we explore the role of discourse structure in document-level machine translation. We introduce a more sound paragraph-to-paragraph translation mode than the several surrounding sentences or fixed length of texts used in previous studies. To better take advantage of the RST parsing tree, we propose the RST-Att model with a multi-granularity attention mechanism depending on the tree structure. The experiment results prove the superiority of our method, and further evaluation indicates that both the guidance of discourse structure and more levels of granularity contribute to the improvement. And the more effective utilization of RST labels for DocMT is left to future research.

## 7 Limitations

Some limitations exist in our research and may be able to be solved in future research. Firstly, current studies on discourse parsing have not achieved great success due to insufficient labeled data. However, the recent improvements in this domain are significant, and the parsing model we employed will directly benefit from the advancement of pre-trained language models. We also believe that better discourse understanding and dealing with multilingual issues would be more beneficial, and we intend to dig into this field in future research and attract more research attention. On the other hand, despite the progressive attention module reducing the calculation, our model does not significantly optimize the cost of time consumed in training. Since we utilize mBART25 which contains quite a few parameters, the process of finetuning may have a minor environmental impact. In future research, we will also further explore how to take better advantage of RST labels in the parsing tree, which may be useful for document-level machine translation.

## 8 Acknowledgements

This work was supported by National Key R&D Program of China (2021YFF0901502), National Science Foundation of China (No. 62161160339), State Key Laboratory of Media Convergence Production Technology and Systems and Key Laboratory of Science, Technology and Standard in Press Industry (Key Laboratory of Intelligent Press Media Technology). We appreciate the anonymous reviewers for their helpful comments, and everyone who has provided assistance in this work. Xiaojun Wan is the corresponding author.

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

## A    Detailed Experiment Settings

All of the datasets and pretrained language models were obtained from publicly available sources under the MIT license, which can be used in academic research. We conduct our experiments on mBART25 (Liu et al., 2020), which involves 12 layers, 16 heads, 1024-dimension outputs, and an intermediate size of 4096 in both the encoder and decoder. The hyperparameters in our work were determined according to the performance on the valid set during the grid search, which started from the hyperparameters in Liu et al. (2020). We use 0.3 dropout, 0.2 label smoothing, and the Adam optimizer with a learning rate of 3e-5 for training. The warm-up steps are 2000 on TED and News,

| Model | TED | News | Europarl |
|---|---|---|---|
| ChatGPT | 33.60 | 39.40 | 30.40 |
| RST-Att | 31.10 | 32.28 | 34.55 |

Table 5: The comparison on the three prominent Doc-MT datasets between ChatGPT and our RST-Att model. The results of ChatGPT come from Wang et al. (2023).

| Model | Training time seconds per epoch | Inference time seconds per test set |
|---|---|---|
| Sent2Sent | 906 | 86 |
| Para2Para | 1055 | 112 |
| RST-Att | 1127 | 115 |

Table 6: Time consumed in the training and inference processes of different models.

and 2500 on Europarl. The beam size during inference is 5, and the BLEU score is calculated in a maximum order of 4 after removing BPE tokens.

## B    Discussion with LLM

There have been some studies assessing the performance of LLMs on prominent Doc-MT datasets, using the prompt that translates the entire document at once. The results of the popular LLM ChatGPT compared to our RST-Att model are shown in Table 5 with the d-BLEU metric. ChatGPT outperforms our RST-Att model on the TED and News datasets, while it lags behind on the Eurparl dataset. We speculate it may be due to the massive training data of ChatGPT. And the domain of Europarl is not as commonly used compared to the TED and News datasets. Furthermore, we cannot verify whether these datasets, particularly the test sets, have already been seen by ChatGPT, as they are publicly available and have been released early. Therefore, it is debatable whether these results are comparable. Recent research (Golchin and Surdeanu, 2023) has also focused on this problem and highlighted concerns regarding data contamination in the test sets of downstream tasks.

## C    Time Efficiency

We have compared the translation efficiency of the training and inference processes with the same settings, such as GPUs, batch size, etc. In Table 6, we present the relevant results of our RST-Att model with Sent2Sent and Para2Para baselines on the NC-2016 dataset, and the performance on other datasets is similar. The training time consumed by

Para2Para2 is about 1.16 times that of Sent2Sent, and the inference time is about 1.3 times. We suppose it is because a paragraph is considerably longer than a sentence, and the computational complexity of the transformer framework is quadratic with respect to input length. Our RST-Att model also adopts paragraph-to-paragraph translation and does not introduce new training parameters. But it requires some computation for multi-granularity attention organization, so its time consumed is a bit greater than the Para2Para baseline. Considering the theoretically lower computation of the attention module, the further efficiency optimization of the implementation of our method is left to future work.