# OpenReview forum: "Exploring Discourse Structure in Document-level Machine Translation"
_EMNLP/2023/Conference — EMNLP 2023 Main_

### Official Review · Reviewer_JdmR · 2023-08-03

**Soundness:** 3

**Excitement:**

3: Ambivalent: It has merits (e.g., it reports state-of-the-art results, the idea is nice), but there are key weaknesses (e.g., it describes incremental work), and it can significantly benefit from another round of revision. However, I won't object to accepting it if my co-reviewers champion it.

**Paper Topic And Main Contributions:**

This paper deals with the task of document-level NMT.
The authors proposed a pipeline, where the paragraphs are first segmented by the TextTiling algorithm, then RST parsing is used to predict the EDUs and construct the RST trees, a linearization step is then used to convert the tree into a 1-dimensional sequence with special syntax to preserve the tree information, and finally a multi-granularity attention is used where early encoder layers are constrained to process local information and later encoder layers are allowed to access global information.

Overall, I think the method is interesting but the experiments are lacking.
While the authors showed decent document Bleu score improvements compared to previous work, I feel several baselines are missing to justify the improvements.

EDIT: After reading the author's rebuttal, many of my concerns are resolved. I feel if the authors can incorporate the points in the rebutal into the next version of the paper, e.g. including the clearly the pretraining data conditions, the contrastive test set performances, and detailed discussions about how a simple separator approach is worse than their method, the paper can be of better quality. Considering this and encouraged by the authors' ethusiasm in responding to questions and willingness to improve the paper, I decided to improve my soundness rating from 2 to 3.

**Questions For The Authors:**

It is quite common to also look at performance on some dedicated contrastive test sets to further justify the improvements from utilizing document-level context. Have you checked those? How does your method compare to the baseline?

**Reasons To Accept:**

The paper is well-written and easy to follow.

The proposed method to segment -> parse -> linearize -> attend is interesting and linguistically well-motivated.

The best setup (RST-Att) outperforms previous work on document Bleu scores.

**Reasons To Reject:**

The experiment section needs to be improved.

For example, the proposed method is finetuned from pretrained BART - a natural question to ask is, in Table 2, have all models seen the same amount of data? If not, how different? Adding a column about this could help the reader to better understand the data conditions. Without it, one can argue that the improvements mainly come from the extra data seen during pre-training.

While the linearization and multi-gran.-atten steps are interesting, I feel simple baselines are missing to justify the method. For instance, in Figure 3, after the EDU segmentation step, one can simply insert artificial <sep> tokens to denote the boundaries. Including such a baseline can help better justify the RST parsing tree step and the linearization step. Note that the multi-gran.-atten can still be applied in such a case.

Basically, my impression after reading the paper was, "hmm, the idea is interesting, but I just saw a bunch of d-Bleu scores that are better than some other work, where does the improvement really come from?" I feel carefully redesigning some baselines/ablations could significantly improve the clarity of the paper.

**Reproducibility:**

4: Could mostly reproduce the results, but there may be some variation because of sample variance or minor variations in their interpretation of the protocol or method.

**Reviewer Confidence:**

3: Pretty sure, but there's a chance I missed something. Although I have a good feel for this area in general, I did not carefully check the paper's details, e.g., the math, experimental design, or novelty.

**Typos Grammar Style And Presentation Improvements:**

My biggest concern is Table 2, while it is not uncommon to see tables like this nowadays, I have to say that as a reader, little can be learned.
Each row is behind a codename, which, unless one is super familiar with the work, tells not much about the actual data/model/training/search setup.
As a result, apart from "this work is about +1 absolute d-Bleu better", not much be can summarized from the table.
I would prefer if relevant information like "pretrained or not?", "training data size", "training criterion", "attention variant", etc. can be listed as additional columns to aid the understanding of the reader.

---

> ### Author Rebuttal · Authors · 2023-08-29
>
> Thank you for your time and efforts, as well as the insightful comments, and we address your concerns below:
>
> **Q1: In Table 2, have all models seen the same amount of data? If not, how different?**
>
> Thank you for your constructive suggestions. In Table 2, we have indicated the pretrained language model (PLM) in parentheses if the corresponding work used it. And other works did not employ PLMs, or their methods could not be directly applied to PLMs. Moreover, all works in Table 2 were trained on the same document-level translation dataset without additional corpora. Given the prevalent use of PLMs in current NLP research and the good generalization of our method, we conduct our experiments directly based on mBART. The setting aligns with the state-of-the-art model G-Transformer in previous works, which serves as our primary comparison object.
>
> On the other hand, we would like to argue that the improvements of our method do not mainly come from extra pre-training data, such as knowledge in mBART. We have supplemented the results of the traditional sub-document translation baseline. As described in our introduction section, some research has pointed out that the standard document-to-document translation mode is prone to training failures. Therefore, many studies adopted the method of splitting each document into sub-documents based on a specified maximum length for better training. In the table below, we provide the comparison between the sub-document (SDoc2SDoc) baseline, our Para2Para baseline, and G-Transformer, all fine-tuned on mBART.
>
> | Model fine-tuned on mBART |  TED  | News  | Europarl |
> | ------------------------- | :---: | :---: | :------: |
> | SDoc2SDoc Baseline [1]    | 28.29 | 30.49 |  34.00   |
> | G-Transformer [1]         | 30.03 | 31.71 |  34.31   |
> | Para2Para Baseline        | 30.35 | 31.43 |  34.19   |
>
> Our Para2Para baseline outperforms the SDoc2SDoc baseline with the same PLM, and the latter actually performs worse than many existing methods on both the TED and Europarl datasets. It can be attributed to the fact that the simple SDoc2SDoc mode, without any other improvements, may arbitrarily ignore the interactions between strongly related sentences. In contrast, our proposed paragraph-to-paragraph method is more reasonable and is expected to make better use of contextual information. The results also prove the considerable improvements from our new paragraph-to-paragraph translation mode itself. Furthermore, we introduced discourse information and proposed our RST-Att model, which has achieved better translation performance. Finally, we agree that the description in Table 2 is not sufficiently clear. If our paper is accepted, we will improve Table 2 with more information on data conditions as well as the relevant discussions above.
>
> **Q2: While the linearization and multi-gran.-atten steps are interesting, I feel simple baselines are missing to justify the method.**
>
> Thank you for your constructive suggestions. In fact, we have experimented before with the baseline you mentioned, which involves adding <sep> tokens to denote EDU segmentation boundaries. However, the performance was basically the same as not adding these tokens, namely the Para2Para Baseline in Table 2. Furthermore, we have also attempted to initialize these segmentation tokens with the semantic representations of the corresponding EDUs to enhance relevant information, but this did not lead to improvements either. As a result, we did not include these baselines in our final paper. We suppose that these simple methods may not be sufficient to help the model leverage discourse information and improve translation performance.
>
> Based on the above attempts, our Para2Para baseline, together with the further RST-Seq setting presented in Table 2, can justify the effectiveness of the linearization step. Additionally, our ablation studies in Table 3 prove that the introduction of the RST parsing tree and the involved multi-granularity attention mechanism indeed further improve translation performance. Moreover, we agree that more baselines and analyses can enhance the clarity of our paper. If our paper is accepted, we will supplement the discussions regarding our attempts mentioned above and carefully improve our experiment section.
>
> **Q3: It is quite common to also look at performance on some dedicated contrastive test sets to further justify the improvements from utilizing document-level context. Have you checked those? How does your method compare to the baseline?**
>
> Thank you for your constructive suggestions. We have supplemented an additional evaluation experiment based on the most frequently used contrastive test sets [2] in existing works. They were designed for targeted evaluation of several discourse phenomena: deixis, ellipsis (two test sets for VP ellipsis and morphological inflection), and lexical cohesion. Each test instance consists of a true example and several contrastive translations that differ from the true one only in the considered discourse aspect. The translation system needs to select the one that it considers to be true, which is evaluated by accuracy. We compared the results of existing research on document-level machine translation and popular large language models (LLMs) with our RST-Att model, as shown in the table below.
>
> | Model                  |  deixis  | lexical cohesion | ellipsis.infl | ellipsis.VP |
> | ---------------------- | :------: | :--------------: | :-----------: | :---------: |
> | Sent2Sent Baseline [2] |   50.0   |       45.9       |     53.0      |    28.9     |
> | Concat Baseline [2]    |   83.5   |       47.5       |     76.6      |    76.2     |
> | CADec [2]              |   81.6   |       58.1       |     72.2      |    80.0     |
> | LSTM-Transformer [3]   | **91.0** |       46.9       |     82.2      |    78.2     |
> | MR-Doc2Doc [4]         |   64.7   |       46.3       |     65.9      |    53.0     |
> | G-Transformer [1]      |   89.9   |        /         |     84.8      |    82.4     |
> | ChatGPT [5]            |   57.9   |       44.4       |     75.0      |    71.6     |
> | GPT4 [5]               |   85.9   |       72.4       |     69.8      |    81.4     |
> | RST-Att (ours)         |   87.2   |     **81.7**     |   **85.2**    |  **87.2**   |
>
> Although the time of rebuttal period is limited and our experiment is possibly not optimized to the best extent, our RST-Att model has achieved superior performance. It demonstrates the highest accuracy in three aspects: lexical cohesion, VP ellipsis, and morphological inflection, while having lower performance in deixis compared to SOTA. And the improvement in terms of lexical cohesion over previous methods is notably significant. Moreover, our approach outperforms LLMs comprehensively, including ChatGPT and GPT-4. It indicates the enhanced ability of our model to adapt to contextual information and handle discourse phenomena, which proves the promising way to introduce discourse information into document-level machine translation. If our paper is accepted, we will add the evaluation experiment mentioned above along with further relevant analysis to our paper.
>
> **Q4: Typos Grammar Style And Presentation Improvements:**
>
> Thank you for your constructive suggestions. We apologize for our oversight, and we agree that adding more details about the previous works in our Table 2 can help provide readers with an overview and a more straightforward comparison. Due to the space limitation, it is challenging to incorporate additional columns containing so much relevant information into the table of experimental results. Therefore, we consider adding an extra table in the appendix to introduce these detailed setups. If our work is accepted, we will carefully address these improvements.
>
> **References:**
>
> [1] Guangsheng Bao, Yue Zhang, Zhiyang Teng, Boxing Chen, and Weihua Luo. 2021. G-transformer for document-level machine translation. In *Proceedings of the 59th Annual Meeting of the Association for Computational Linguistics and the 11th International Joint Conference on Natural Language Processing (Volume 1: Long Papers)*, pages 3442–3455.
>
> [2] Elena Voita, Rico Sennrich, and Ivan Titov. 2019b. When a good translation is wrong in context: Context-aware machine translation improves on deixis, ellipsis, and lexical cohesion. In *Proceedings of the 57th Annual Meeting of the Association for Computational Linguistics*, pages 1198–1212, Florence, Italy. Association for Computational Linguistics.
>
> [3] Pei Zhang, Boxing Chen, Niyu Ge, and Kai Fan. 2020. Long-short term masking transformer: A simple but effective baseline for document-level neural machine translation. In *Proceedings of the 2020 Conference on Empirical Methods in Natural Language Processing (EMNLP)*, pages 1081–1087, Online. Association for Computational Linguistics.
>
> [4] Zewei Sun, Mingxuan Wang, Hao Zhou, Chengqi Zhao, Shujian Huang, Jiajun Chen, and Lei Li. 2022. Rethinking document-level neural machine translation. In *Findings of the Association for Computational Linguistics: ACL 2022*, pages 3537–3548.
>
> [5] Longyue Wang, Chenyang Lyu, Tianbo Ji, Zhirui Zhang, Dian Yu, Shuming Shi, Zhaopeng Tu. 2023. Document-level machine translation with large language models[J]. *arXiv preprint arXiv:2304.02210*.

---

### Official Review · Reviewer_EkAb · 2023-08-05

**Soundness:** 4

**Excitement:**

4: Strong: This paper deepens the understanding of some phenomenon or lowers the barriers to an existing research direction.

**Paper Topic And Main Contributions:**

This paper presents a new paragraph-to-paragraph model for document machine translation. This method firstly uses TextTiling to splits the document into paragraphs and parsers each paragraph into an RST tree. The RST tree is further linearized into a sequence to replace the original text input for the seq2seq model. The author further proposes a method using attention to better utilize the According to the experiment, this method achieves good results, especially the RST-Att method, which achieves sota performance when this work is done.

In general, the idea in this paper is well-motivated and achieves good performance. But this work still needs more analysis and experiments to figure out where the performance comes from.

**Questions For The Authors:**

-Lines 188-191: The authors mentioned, "TextTiling method has a more reasonable distribution, and the variance of the numbers of tokens and sentences is smaller." I am not confident that this claim is established since the variance of the number of tokens and sentences is an inherent feature of the language, which could be large. I hope the authors can explain more about this claim.

**Reasons To Accept:**

-This idea of splitting documents into paragraphs and using an RST tree in machine translation is interesting and inspiring.

-This paper proposes a new mode for document translation that splits documents into paragraphs first and then translates paragraphs. This paper also proposes new models to utilize discourse information through attention better to achieve better performance and reduce computation complexity. According to experiments, this method performs well by outperforming the Sota method when this work is done.

**Reasons To Reject:**

-The notation for the "Para2Para Baseline (BART)" lacks a formal definition. If I understand correctly, this baseline means the method that performs paragraph-to-paragraph translation but without using RST tree information. It would be better if the authors could add a definition for this notation.
-If my understanding of the first point about "Para2Para Baseline (BART)" is correct, it indicates the baseline could already achieve great performance (almost outperforming most previous works). I can not find enough support analysis about the oblation study about where this improvement comes from. It would be great if the authors could add more analysis.

**Reproducibility:**

4: Could mostly reproduce the results, but there may be some variation because of sample variance or minor variations in their interpretation of the protocol or method.

**Reviewer Confidence:**

4: Quite sure. I tried to check the important points carefully. It's unlikely, though conceivable, that I missed something that should affect my ratings.

---

> ### Author Rebuttal · Authors · 2023-08-29
>
> Thank you for your time and efforts, as well as the insightful comments, and we address your concerns below:
>
> **Q1: The notation for the "Para2Para Baseline (BART)" lacks a formal definition.**
>
> Thank you for your constructive suggestions. We apologize for the lack of a formal definition for this notation, and your understanding is correct. The corresponding description in our paper can be found in line 375. And there was no utilization of RST tree information or modification made to the model architecture in this baseline setting. Instead, it simply performed paragraph-to-paragraph translation based on the paragraphs we had previously segmented. If our paper is accepted, we will improve and clarify the definition of this notation.
>
> **Q2: I can not find enough support analysis about the ablation study about where the improvement of Para2Para Baseline (BART) comes from. It would be great if the authors could add more analysis.**
>
> Thank you for your constructive suggestions. We believe that the improvement of Para2Para Baseline mainly comes from the paragraph translation mode we proposed. For better illustration, we have supplemented the results of the traditional sub-document translation baseline. As described in our introduction section, some studies have pointed out that the standard document-to-document translation mode is prone to training failures. Therefore, many studies adopted the method of splitting each document into sub-documents based on a specified maximum length for better training. In the table below, we provide the comparison between the sub-document (SDoc2SDoc) baseline, our Para2Para baseline, and the previous state-of-the-art model G-Transformer, all fine-tuned on mBART.
>
> | Model fine-tuned on mBART |  TED  | News  | Europarl |
> | ------------------------- | :---: | :---: | :------: |
> | SDoc2SDoc Baseline [1]    | 28.29 | 30.49 |  34.00   |
> | G-Transformer [1]         | 30.03 | 31.71 |  34.31   |
> | Para2Para Baseline        | 30.35 | 31.43 |  34.19   |
>
> Our Para2Para baseline outperforms the traditional SDoc2SDoc baseline, and the latter actually performs worse than many existing methods on both the TED and Europarl datasets. It can be attributed to the fact that the simple SDoc2SDoc mode, without any other improvements, may arbitrarily ignore the interactions between strongly related sentences. In contrast, our proposed paragraph-to-paragraph method is more reasonable and is expected to make better use of contextual information. This is because our paragraphs are segmented under consideration of the internal correlations of sentences in the paragraph. The results also prove the effectiveness of our new translation mode.
>
> Moreover, while the Para2Para baseline outperforms G-Transformer on the TED dataset, it is inferior on the other two datasets. For further improvements, we introduced discourse information and proposed our RST-Att model, which has achieved better translation performance. If our paper is accepted, we will add more relevant analysis as above.
>
> **Q3: Lines 188-191: The authors mentioned, "TextTiling method has a more reasonable distribution, and the variance of the numbers of tokens and sentences is smaller." I am not confident that this claim is established since the variance of the number of tokens and sentences is an inherent feature of the language, which could be large. I hope the authors can explain more about this claim.**
>
> Thank you for your concerns. We apologize that our expression here is not quite accurate. And we agree that different paragraphs should have certain differences in terms of word and sentence counts. What we intended to convey is that, when comparing the two segmentation methods, the TextTiling tool yields a more reasonable distribution of paragraph lengths, concentrating in a moderate number of sentences (4–7 sentences), as shown in Figures 1 and 2. In contrast, the other method results in a large number of short segments (1–3 sentences), which somewhat deviate from the concept of discourse and are not suitable for discourse parsing. If our paper is accepted, we will modify this statement as follows: "The paragraphs from the TextTiling method have a more reasonable distribution, and most of them contain moderate numbers of tokens and sentences."
>
> **Q4: Supplementary evaluation experiments and analysis of discourse phenomena.**
>
> We have supplemented an additional evaluation experiment based on the most frequently used contrastive test sets [2] in existing works. They were designed for targeted evaluation of several discourse phenomena: deixis, ellipsis (two test sets for VP ellipsis and morphological inflection), and lexical cohesion. Each test instance consists of a true example and several contrastive translations that differ from the true one only in the considered discourse aspect. The translation system needs to select the one that it considers to be true, which is evaluated by accuracy. We compared the results of existing research on document-level machine translation and popular large language models (LLMs) with our RST-Att model, as shown in the table below.
>
> | Model                  |  deixis  | lexical cohesion | ellipsis.infl | ellipsis.VP |
> | ---------------------- | :------: | :--------------: | :-----------: | :---------: |
> | Sent2Sent Baseline [2] |   50.0   |       45.9       |     53.0      |    28.9     |
> | Concat Baseline [2]    |   83.5   |       47.5       |     76.6      |    76.2     |
> | CADec [2]              |   81.6   |       58.1       |     72.2      |    80.0     |
> | LSTM-Transformer [3]   | **91.0** |       46.9       |     82.2      |    78.2     |
> | MR-Doc2Doc [4]         |   64.7   |       46.3       |     65.9      |    53.0     |
> | G-Transformer [1]      |   89.9   |        /         |     84.8      |    82.4     |
> | ChatGPT [5]            |   57.9   |       44.4       |     75.0      |    71.6     |
> | GPT4 [5]               |   85.9   |       72.4       |     69.8      |    81.4     |
> | RST-Att (ours)         |   87.2   |     **81.7**     |   **85.2**    |  **87.2**   |
>
> Although the time of rebuttal period is limited and our experiment is possibly not optimized to the best extent, our RST-Att model has achieved superior performance. It demonstrates the highest accuracy in three aspects: lexical cohesion, VP ellipsis, and morphological inflection, while having lower performance in deixis compared to SOTA. And the improvement in terms of lexical cohesion over previous methods is notably significant. Moreover, our approach outperforms LLMs comprehensively, including ChatGPT and GPT-4. It indicates the enhanced ability of our model to adapt to contextual information and handle discourse phenomena, which proves the promising way to introduce discourse information into document-level machine translation. If our paper is accepted, we will add the evaluation experiment mentioned above along with further relevant analysis to our paper.
>
> **References:**
>
> [1] Guangsheng Bao, Yue Zhang, Zhiyang Teng, Boxing Chen, and Weihua Luo. 2021. G-transformer for document-level machine translation. In *Proceedings of the 59th Annual Meeting of the Association for Computational Linguistics and the 11th International Joint Conference on Natural Language Processing (Volume 1: Long Papers)*, pages 3442–3455.
>
> [2] Elena Voita, Rico Sennrich, and Ivan Titov. 2019b. When a good translation is wrong in context: Context-aware machine translation improves on deixis, ellipsis, and lexical cohesion. In *Proceedings of the 57th Annual Meeting of the Association for Computational Linguistics*, pages 1198–1212, Florence, Italy. Association for Computational Linguistics.
>
> [3] Pei Zhang, Boxing Chen, Niyu Ge, and Kai Fan. 2020. Long-short term masking transformer: A simple but effective baseline for document-level neural machine translation. In *Proceedings of the 2020 Conference on Empirical Methods in Natural Language Processing (EMNLP)*, pages 1081–1087, Online. Association for Computational Linguistics.
>
> [4] Zewei Sun, Mingxuan Wang, Hao Zhou, Chengqi Zhao, Shujian Huang, Jiajun Chen, and Lei Li. 2022. Rethinking document-level neural machine translation. In *Findings of the Association for Computational Linguistics: ACL 2022*, pages 3537–3548.
>
> [5] Longyue Wang, Chenyang Lyu, Tianbo Ji, Zhirui Zhang, Dian Yu, Shuming Shi, Zhaopeng Tu. 2023. Document-level machine translation with large language models[J]. *arXiv preprint arXiv:2304.02210*.

---

### Official Review · Reviewer_JzyP · 2023-08-08

**Soundness:** 3

**Excitement:**

3: Ambivalent: It has merits (e.g., it reports state-of-the-art results, the idea is nice), but there are key weaknesses (e.g., it describes incremental work), and it can significantly benefit from another round of revision. However, I won't object to accepting it if my co-reviewers champion it.

**Paper Topic And Main Contributions:**

This paper explores utilizing discourse structure for document-level neural machine translation (DocMT). The authors introduce a paragraph-to-paragraph translation approach and generate paragraph segmentation using TextTiling. They train an end-to-end RST parser to predict discourse structure and propose several methods to incorporate it into DocMT, including integrating the linearized RST sequence (RST-Seq) and an RST tree-based multi-granularity attention mechanism (RST-Att). Experiments on TED, News, and Europarl datasets show their RST-Att model achieves the best results, outperforming prior work on DocMT.

**Main contributions:**

1. Introduces a more principled paragraph-level translation approach compared to prior segmentation methods in DocMT.
2. Presents several concrete attempts to utilize predicted RST structure, with ablation studies showing the benefits of RST-specific guidance and multiple granularity levels.


**Reasons To Accept:**

Same as main contributions

**Reasons To Reject:**

1. Relies entirely on automatic RST parsing, which remains error-prone. Better discourse understanding would likely improve benefits.
2. It seems that such RST parsers are mainly available for English (as such, the submission also mainly conducted evaluation from English to German), making it hard to generalize to other languages.
2. Using external TextTiling tool may limit end-to-end learning of document boundaries.
3. Limited linguistic analysis of what discourse phenomena the model has learned to handle.

**Reproducibility:**

2: Would be hard pressed to reproduce the results. The contribution depends on data that are simply not available outside the author's institution or consortium; not enough details are provided.

**Reviewer Confidence:**

3: Pretty sure, but there's a chance I missed something. Although I have a good feel for this area in general, I did not carefully check the paper's details, e.g., the math, experimental design, or novelty.

---

> ### Author Rebuttal · Authors · 2023-08-29
>
> Thank you for your time and efforts, as well as the insightful comments, and we address your concerns below:
>
> **Q1: Relies entirely on automatic RST parsing, which remains error-prone. Better discourse understanding would likely improve benefits.**
>
> Thank you for your constructive suggestions. The incorporation of discourse information into document-level machine translation is the motivation and novelty of our work. And we primarily considered the RST discourse structure, so it is indeed necessary to rely on automatic RST parsing methods. In fact, the current performance of RST parsing is not unsatisfactory, and the parsing model we have trained demonstrates an F1 score of over 88% on Span in our evaluation.
>
> Furthermore, our approach has achieved improvements with our obtained RST discourse structural information. Given the possibility of further advancement on the RST parser (e.g., fine-tuned on powerful large language models (LLMs)), our work shows a promising direction to promote document-level translation with discourse parsing. And our approach can directly benefit from the corresponding advancements. We also agree that other better discourse understanding would likely improve benefits, and we intend to dig into this field in future research and attract more research attention. If our paper is accepted, we will add relevant explanations as above.
>
> **Q2: It seems that such RST parsers are mainly available for English (as such, the submission also mainly conducted evaluation from English to German), making it hard to generalize to other languages.**
>
> Thank you for your concerns. The three English-to-German datasets in our experiments are the most prominent and most commonly used for document-level machine translation tasks, consistent with many existing works. Moreover, we agree that multilingual issues remain a challenge in the field of discourse parsing, primarily due to the scarcity of well-annotated multilingual corpora. However, with the advancement of cross-lingual discourse parsing research, including the use of LLMs in data augmentation, we believe that relevant parsing tools will soon be available. And they can simliarly be integrated into our approach, thereby improving document-level translation for other languages. We also consider the aforementioned multilingual issues as potential directions for future research.
>
> **Q3: Using external TextTiling tool may limit end-to-end learning of document boundaries.**
>
> Our method does not involve changing the original document boundaries. Instead, we use the TextTiling tool to split each document into several paragraphs with strong internal correlations. In fact, previous studies on document-level machine translation did not conduct end-to-end document boundary learning, namely standard document-to-document translation. This approach has been shown in some works to be prone to training failures. So many studies adopted a method of splitting each document into sub-documents based on a specified maximum length for better training.
>
> As we pointed out in the introduction section, we think this common mode may arbitrarily ignore the interactions between strongly related sentences in the document. In contrast, our proposed paragraph-to-paragraph method is more reasonable and is expected to make better use of contextual information. We provide a comparison between a sub-document (SDoc2SDoc) translation baseline and our Para2Para baseline, both without changes in model frameworks, as shown in the table below. Our paragraph-to-paragraph translation mode itself demonstrates better performance than the traditional SDoc2SDoc baseline before introducing discourse information, which proves the effectiveness of our new translation mode.
>
> | Model                  |  TED  | News  | Europarl |
> | ---------------------- | :---: | :---: | :------: |
> | SDoc2SDoc Baseline [1] | 28.29 | 30.49 |  34.00   |
> | Para2Para Baseline     | 30.35 | 31.43 |  34.19   |
>
> **Q4: Limited linguistic analysis of what discourse phenomena the model has learned to handle.**
>
> Thank you for your constructive suggestions. We have supplemented an additional evaluation experiment based on the most frequently used contrastive test sets [2] in existing works. They were designed for targeted evaluation of several discourse phenomena: deixis, ellipsis (two test sets for VP ellipsis and morphological inflection), and lexical cohesion. Each test instance consists of a true example and several contrastive translations that differ from the true one only in the considered discourse aspect. The translation system needs to select the one that it considers to be true, which is evaluated by accuracy. We compared the results of existing research on document-level machine translation and popular large language models (LLMs) with our RST-Att model, as shown in the table below.
>
> | Model                  |  deixis  | lexical cohesion | ellipsis.infl | ellipsis.VP |
> | ---------------------- | :------: | :--------------: | :-----------: | :---------: |
> | Sent2Sent Baseline [2] |   50.0   |       45.9       |     53.0      |    28.9     |
> | Concat Baseline [2]    |   83.5   |       47.5       |     76.6      |    76.2     |
> | CADec [2]              |   81.6   |       58.1       |     72.2      |    80.0     |
> | LSTM-Transformer [3]   | **91.0** |       46.9       |     82.2      |    78.2     |
> | MR-Doc2Doc [4]         |   64.7   |       46.3       |     65.9      |    53.0     |
> | G-Transformer [1]      |   89.9   |        /         |     84.8      |    82.4     |
> | ChatGPT [5]            |   57.9   |       44.4       |     75.0      |    71.6     |
> | GPT4 [5]               |   85.9   |       72.4       |     69.8      |    81.4     |
> | RST-Att (ours)         |   87.2   |     **81.7**     |   **85.2**    |  **87.2**   |
>
> Although the time of rebuttal period is limited and our experiment is possibly not optimized to the best extent, our RST-Att model has achieved superior performance. It demonstrates the highest accuracy in three aspects: lexical cohesion, VP ellipsis, and morphological inflection, while having lower performance in deixis compared to SOTA. And the improvement in terms of lexical cohesion over previous methods is notably significant. Moreover, our approach outperforms LLMs comprehensively, including ChatGPT and GPT-4. It indicates the enhanced ability of our model to adapt to contextual information and handle discourse phenomena, which proves the promising way to introduce discourse information into document-level machine translation. If our paper is accepted, we will add the evaluation experiment mentioned above along with further relevant analysis to our paper.
>
> **Q5: About reproducibility.**
>
> In fact, the datasets involved in our work are common and widely used in existing research. For RST parsing, we will provide the corresponding training codes for convenient reproducing. We will also release our trained parsing model, which can be directly used. Regarding document-level translation, we will release the discourse parsing information of the relevant datasets, along with codes for our models and methods, to ensure the reproducibility of our work.
>
> **References:**
>
> [1] Guangsheng Bao, Yue Zhang, Zhiyang Teng, Boxing Chen, and Weihua Luo. 2021. G-transformer for document-level machine translation. In *Proceedings of the 59th Annual Meeting of the Association for Computational Linguistics and the 11th International Joint Conference on Natural Language Processing (Volume 1: Long Papers)*, pages 3442–3455.
>
> [2] Elena Voita, Rico Sennrich, and Ivan Titov. 2019b. When a good translation is wrong in context: Context-aware machine translation improves on deixis, ellipsis, and lexical cohesion. In *Proceedings of the 57th Annual Meeting of the Association for Computational Linguistics*, pages 1198–1212, Florence, Italy. Association for Computational Linguistics.
>
> [3] Pei Zhang, Boxing Chen, Niyu Ge, and Kai Fan. 2020. Long-short term masking transformer: A simple but effective baseline for document-level neural machine translation. In *Proceedings of the 2020 Conference on Empirical Methods in Natural Language Processing (EMNLP)*, pages 1081–1087, Online. Association for Computational Linguistics.
>
> [4] Zewei Sun, Mingxuan Wang, Hao Zhou, Chengqi Zhao, Shujian Huang, Jiajun Chen, and Lei Li. 2022. Rethinking document-level neural machine translation. In *Findings of the Association for Computational Linguistics: ACL 2022*, pages 3537–3548.
>
> [5] Longyue Wang, Chenyang Lyu, Tianbo Ji, Zhirui Zhang, Dian Yu, Shuming Shi, Zhaopeng Tu. 2023. Document-level machine translation with large language models[J]. *arXiv preprint arXiv:2304.02210*.

---

### Official Review · Reviewer_2yPU · 2023-08-14

**Paper Topic And Main Contributions:** 1. Introducing a robust paragraph-to-…
**Soundness:** 4

**Excitement:**

3: Ambivalent: It has merits (e.g., it reports state-of-the-art results, the idea is nice), but there are key weaknesses (e.g., it describes incremental work), and it can significantly benefit from another round of revision. However, I won't object to accepting it if my co-reviewers champion it.

**Questions For The Authors:**

1. How much slower is the decoding efficiency compared to translating the entire sentence? It is hoped that the authors can provide some experimental results related to translation efficiency;

2. This is still a traditional approach applied to translation models. How does it compare with the currently prevalent large language models (LLMs)? I would like to see comparative experimental results.

**Reasons To Accept:**

This paper addresses a significant challenge in the field of neural machine translation - the limitation of existing methods for document-level machine translation (DocMT) when processing lengthy texts. The proposed approach, which employs discourse structure analysis to enhance context utilization, is novel and well-motivated. The experimental results demonstrate the superiority of the proposed RST-Att model over existing approaches, showcasing its potential to advance the state-of-the-art in DocMT. The paper's clarity, solid theoretical foundation, and empirical validation make it a valuable contribution to the NLP community.

**Reasons To Reject:**

While the topic of enhancing document-level machine translation through discourse structure analysis is intriguing, the paper lacks empirical validation and practical implementation. The proposed approach and models are described in detail, but there is a need for experimental results on relevant datasets to demonstrate the efficacy of the approach. Additionally, the paper could benefit from a more thorough comparison with existing methods in terms of performance and computational efficiency. As it stands, the manuscript lacks the necessary depth to make a meaningful contribution to the field.

**Reproducibility:**

4: Could mostly reproduce the results, but there may be some variation because of sample variance or minor variations in their interpretation of the protocol or method.

**Reviewer Confidence:**

4: Quite sure. I tried to check the important points carefully. It's unlikely, though conceivable, that I missed something that should affect my ratings.

---

> ### Author Rebuttal · Authors · 2023-08-29
>
> Thank you for your time and efforts, as well as the insightful comments, and we address your concerns below:
>
> **Q1:  There is a need for experimental results on relevant datasets to demonstrate the efficacy of the approach.**
>
> Thank you for your constructive suggestions. In fact, we designed several baseline experiments to compare with our final RST-Att model. And we also conducted ablation studies on the involved multi-granularity attention mechanism to demonstrate the efficacy of our approach to introducing RST discourse structure. The relevant results are presented in Table 2 and Table 3 of our paper.
>
> Furthermore, we have supplemented an additional evaluation experiment based on the most frequently used contrastive test sets [1] in existing works. They were designed for targeted evaluation of several discourse phenomena: deixis, ellipsis (two test sets for VP ellipsis and morphological inflection), and lexical cohesion. Each test instance consists of a true example and several contrastive translations that differ from the true one only in the considered discourse aspect. The translation system needs to select the one that it considers to be true, which is evaluated by accuracy. We compared the results of existing research on document-level machine translation and popular large language models (LLMs) with our RST-Att model, as shown in the table below.
>
> | Model                  |  deixis  | lexical cohesion | ellipsis.infl | ellipsis.VP |
> | ---------------------- | :------: | :--------------: | :-----------: | :---------: |
> | Sent2Sent Baseline [1] |   50.0   |       45.9       |     53.0      |    28.9     |
> | Concat Baseline [1]    |   83.5   |       47.5       |     76.6      |    76.2     |
> | CADec [1]              |   81.6   |       58.1       |     72.2      |    80.0     |
> | LSTM-Transformer [2]   | **91.0** |       46.9       |     82.2      |    78.2     |
> | MR-Doc2Doc [3]         |   64.7   |       46.3       |     65.9      |    53.0     |
> | G-Transformer [4]      |   89.9   |        /         |     84.8      |    82.4     |
> | ChatGPT [5]            |   57.9   |       44.4       |     75.0      |    71.6     |
> | GPT4 [5]               |   85.9   |       72.4       |     69.8      |    81.4     |
> | RST-Att (ours)         |   87.2   |     **81.7**     |   **85.2**    |  **87.2**   |
>
> Although the time of rebuttal period is limited and our experiment is possibly not optimized to the best extent, our RST-Att model has achieved superior performance. It demonstrates the highest accuracy in three aspects: lexical cohesion, VP ellipsis, and morphological inflection, while having lower performance in deixis compared to SOTA. And the improvement in terms of lexical cohesion over previous methods is notably significant. Moreover, our approach outperforms LLMs comprehensively, including ChatGPT and GPT-4. It indicates the enhanced ability of our model to adapt to contextual information and handle discourse phenomena, which proves the promising way to introduce discourse information into document-level machine translation. If our paper is accepted, we will add the evaluation experiment mentioned above along with further relevant analysis to our paper.
>
> **Q2: How much slower is the decoding efficiency compared to translating the entire sentence? It is hoped that the authors can provide some experimental results related to translation efficiency.**
>
> Thank you for another significant suggestion that helps to improve the quality of our work. Due to the limited time, we tested the sentence-to-sentence (Sent2Sent) and paragraph-to-paragraph (Para2Para) baselines, along with our RST-Att model. Previous works did not show the results in this regard, and we did not have enough time to reproduce them. We have compared the translation efficiency of the training and inference processes with the same settings, such as a single NVIDIA A40 GPU, the Fairseq framework, batch size, etc. In the table below, we present the relevant results on the NC-2016 dataset, and the performance on other datasets is similar.
>
> | Model              | Training time (seconds per epoch) | Inference time (seconds per epoch) |
> | ------------------ | :-------------------------------: | :--------------------------------: |
> | Sent2Sent Baseline |                906                |                 86                 |
> | Para2Para Baseline |               1055                |                112                 |
> | RST-Att            |               1127                |                115                 |
>
> The training time consumed by the Para2Para2 baseline is about 1.16 times that of the Sent2Sent baseline, and the inference time is about 1.3 times. We suppose it is because a paragraph is considerably longer than a sentence, and the computational complexity of the transformer framework is quadratic with respect to input length. Our RST-Att model also adopts the input of paragraphs and does not introduce new training parameters. But it requires some computation for multi-granularity attention organization, so its time consumed is a bit greater than the Para2Para baseline.
>
> Despite slightly lower translation efficiency, our approach has improved the quality of document-level translation by better utilizing contextual content and discourse information. In addition, our multi-granularity attention mechanism actually can reduce computational overhead, as illustrated in Figure 4. Therefore, there is much room for further efficiency optimization, which was not primarily focused on in our work before. If our paper is accepted, we will add the relevant results and discussions, along with improvements to our model implementation.
>
> **Q3: This is still a traditional approach applied to translation models. How does it compare with the currently prevalent large language models (LLMs)? I would like to see comparative experimental results.**
>
> There have been some studies assessing the performance of LLMs on prominent document-level machine translation datasets, using the prompt that translates the entire document at once. The results of the popular LLM ChatGPT compared to our RST-Att model are as follows, with the d-BLEU metric for evaluation.
>
> | Model          |  TED  | News  | Europarl |
> | -------------- | :---: | :---: | :------: |
> | ChatGPT [5]    | 33.6  | 39.4  |   30.4   |
> | RST-Att (ours) | 31.10 | 32.28 |  34.55   |
>
> ChatGPT outperforms our RST-Att model on the TED and News datasets, while it lags behind on the Eurparl dataset. We speculate it may be due to the massive training data of ChatGPT. In fact, we cannot verify whether these datasets, particularly the test sets, have already been seen by ChatGPT, as they are publicly available and have been released early. Therefore, it is debatable whether these results are comparable. Recent research [6] has also focused on this problem and highlighted concerns regarding data contamination in downstream task test sets.
>
> **References:**
>
> [1] Elena Voita, Rico Sennrich, and Ivan Titov. 2019b. When a good translation is wrong in context: Context-aware machine translation improves on deixis, ellipsis, and lexical cohesion. In *Proceedings of the 57th Annual Meeting of the Association for Computational Linguistics*, pages 1198–1212, Florence, Italy. Association for Computational Linguistics.
>
> [2] Pei Zhang, Boxing Chen, Niyu Ge, and Kai Fan. 2020. Long-short term masking transformer: A simple but effective baseline for document-level neural machine translation. In *Proceedings of the 2020 Conference on Empirical Methods in Natural Language Processing (EMNLP)*, pages 1081–1087, Online. Association for Computational Linguistics.
>
> [3] Zewei Sun, Mingxuan Wang, Hao Zhou, Chengqi Zhao, Shujian Huang, Jiajun Chen, and Lei Li. 2022. Rethinking document-level neural machine translation. In *Findings of the Association for Computational Linguistics: ACL 2022*, pages 3537–3548.
>
> [4] Guangsheng Bao, Yue Zhang, Zhiyang Teng, Boxing Chen, and Weihua Luo. 2021. G-transformer for document-level machine translation. In *Proceedings of the 59th Annual Meeting of the Association for Computational Linguistics and the 11th International Joint Conference on Natural Language Processing (Volume 1: Long Papers)*, pages 3442–3455.
>
> [5] Longyue Wang, Chenyang Lyu, Tianbo Ji, Zhirui Zhang, Dian Yu, Shuming Shi, Zhaopeng Tu. 2023. Document-level machine translation with large language models[J]. *arXiv preprint arXiv:2304.02210*.
>
> [6] Shahriar Golchin and Mihai Surdeanu. 2023. Time Travel in LLMs: Tracing Data Contamination in Large Language Models. *arXiv preprint arXiv:2308.08493*.

---

### Meta-Review · Area_Chair_aFdL · 2023-09-18

**Recommendation:** 4

**Metareview:**

This paper proposes an approach to document-level NMT that uses a rhetorical structure theory (RST) parser to capture discourse information. It explores different methods for adding this information to the encoder, notably a custom attention mechanism that uses the parse to control the context available to different layers. This is shown to outperform a wide range of baselines across different test settings.

Reviewers noted that the paper tackles a significant problem, and found the approach novel,  well-motivated, and interesting, with  positive results from extensive experiments, including ablation studies. The main weakness is the reliance on the RST parser, which is not guaranteed to be free from errors, and which limits source languages to those for which such a parser is available. Some reviewers also felt that analysis specifically linking quality improvements to linguistic aspects of the context would have strengthened the paper.

This is substantial work that represents a very strong baseline for document-level NMT. Although it’s hard to see it as the future of document-level MT - given the pipelined design, specially-modified architecture, and reliance on custom linguistic components - it is nonetheless an interesting point for comparison and analysis.

---

### Decision · Program_Chairs · 2023-10-07

**Decision:**

Accept-Main

**Comment:**

This paper proposes an approach to document-level NMT that uses a rhetorical structure theory (RST) parser to capture discourse information. It explores different methods for adding this information to the encoder, notably a custom attention mechanism that uses the parse to control the context available to different layers. This is shown to outperform a wide range of baselines across different test settings.

Reviewers noted that the paper tackles a significant problem, and found the approach novel,  well-motivated, and interesting, with  positive results from extensive experiments, including ablation studies. The main weakness is the reliance on the RST parser, which is not guaranteed to be free from errors, and which limits source languages to those for which such a parser is available. Some reviewers also felt that analysis specifically linking quality improvements to linguistic aspects of the context would have strengthened the paper.

This is substantial work that represents a very strong baseline for document-level NMT. Although it’s hard to see it as the future of document-level MT - given the pipelined design, specially-modified architecture, and reliance on custom linguistic components - it is nonetheless an interesting point for comparison and analysis.